# tRNA Biology in the Pathogenesis of Diabetes: Role of Genetic and Environmental Factors

**DOI:** 10.3390/ijms22020496

**Published:** 2021-01-06

**Authors:** Maria Nicol Arroyo, Jonathan Alex Green, Miriam Cnop, Mariana Igoillo-Esteve

**Affiliations:** 1ULB Center for Diabetes Research, Université Libre de Bruxelles, 1050 Brussels, Belgium; maria.arroyo@ulb.ac.be (M.N.A.); jonathan.green@ulb.ac.be (J.A.G.); mcnop@ulb.ac.be (M.C.); 2Division of Endocrinology, Erasmus Hospital, Université Libre de Bruxelles, 1050 Brussels, Belgium

**Keywords:** type 2 diabetes, tRNA, tRNA fragments, tRNA modifications, pancreatic β-cells, obesity, insulin resistance

## Abstract

The global rise in type 2 diabetes results from a combination of genetic predisposition with environmental assaults that negatively affect insulin action in peripheral tissues and impair pancreatic β-cell function and survival. Nongenetic heritability of metabolic traits may be an important contributor to the diabetes epidemic. Transfer RNAs (tRNAs) are noncoding RNA molecules that play a crucial role in protein synthesis. tRNAs also have noncanonical functions through which they control a variety of biological processes. Genetic and environmental effects on tRNAs have emerged as novel contributors to the pathogenesis of diabetes. Indeed, altered tRNA aminoacylation, modification, and fragmentation are associated with β-cell failure, obesity, and insulin resistance. Moreover, diet-induced tRNA fragments have been linked with intergenerational inheritance of metabolic traits. Here, we provide a comprehensive review of how perturbations in tRNA biology play a role in the pathogenesis of monogenic and type 2 diabetes.

## 1. Introduction

Type 2 diabetes (T2D) is a complex metabolic disease that accounts for more than 80% of all diabetes cases. Its prevalence is continuously increasing as a result of the worldwide obesity epidemic and sedentary lifestyle [1]. A large number of genetic variants predisposing to T2D have been identified; however, when considered individually, they only modestly increase T2D risk [2,3]. Therefore, the presence of T2D risk alleles in multiple genes together with disease-conductive environmental factors are required for T2D to develop. Central adiposity is considered an important risk factor for T2D since it increases levels of circulating free fatty acids (FFAs), oxidative stress, and inflammatory molecules that induce insulin resistance in peripheral tissues and impair pancreatic β-cell function [4]. Genetic and environmentally mediated changes in transfer RNA (tRNA) metabolism have recently emerged as a novel pathogenic mechanism underlying diabetes through effects on β-cell function and survival, obesity, and insulin resistance [5,6,7]. Here we review recent findings in this area with a special focus on the association between altered tRNA aminoacylation, dysregulated tRNA modifications, and tRNA fragmentation with impaired glucose metabolism

## 2. tRNA Biogenesis, Structure, and Function

tRNAs are small ~73–90 nucleotide-long RNA molecules that work as aminoacid carriers during protein synthesis. They are generated by RNA polymerase III (Pol III) as precursor tRNAs containing 5′ leader and 3′ trailer sequence extensions which are cleaved to generate mature tRNAs [8]. Through the action of highly specific aminoacyl-tRNA synthetases (aaRSs), each tRNA is charged with its corresponding aminoacid that is covalently linked to the 3′ end of the molecule (Figure 1) [9]. Cytosolic tRNAs have a 2D cloverleaf structure that comprises a D-loop, a variable loop, a T loop, and an anticodon loop (Figure 1). The D loop contains one dihydrouridine that is hypothesized to serve as the recognition site for the aaRS [10]. The 3 to 21 nucleotide-long variable loop promotes tRNA stabilization and aaRS recognition, while the T loop, also known as the thymidine, pseudouridine, and cytidine (TΨC) loop, contributes to tRNA folding in the 3D L shape and serves as the ribosome recognition site during the tRNA-ribosome complex formation in protein biosynthesis [11]. The anticodon loop actively participates in translation since it contains the three nucleotides complementary to the codon present in the mRNA [12] (Figure 1).

While only 22 mitochondrial tRNAs (mt-tRNAs) exist that are essential for mitochondrial protein translation [13], more than 260 cytosolic tRNAs constitute the cell- and tissue-specific tRNA pool [14,15,16,17,18]. Indeed, each aminoacid can be transported by a wide number of tRNAs that are classified as isoacceptors or isodecoders. Carrying the same aminoacid, isoacceptors have different anticodon and tRNA body sequences, while isodecoders only differ in the tRNA body sequence [19]. 

## 3. Post-Transcriptional tRNA Modifications 

tRNAs undergo extensive post-transcriptional modifications that are carried out by specific tRNA-modifying enzymes. Over 80 different tRNA modifications have been reported that are present in both cytosolic and mt-tRNAs [18,19]. While cytosolic tRNAs have an average of 13 modified bases, mt-tRNAs are generally less modified with around five modifications per tRNA [19]. The tRNA modifications are distributed all along the molecule and, depending on their localization, impact tRNA stability or function and modulate tRNA fragmentation (see below) [17,19,20,21]. As one of the main functions of tRNA modifications is to improve tRNA stability, the presence of a limited number of modifications in mt-tRNAs contributes to their lower stability and their dependency on unique structural integrity modifications [22]. While both cytosolic and mt-tRNA share some modifications, as is the case of N^6^-isopentenyl modification to adenosine (i^6^A) at position 37, there are unique modifications that are restricted to specific tRNA categories, e.g., taurine modifications that are exclusively seen in mt-tRNAs [23]. Modifications at position 34 (the wobble tRNA position) and position 37 are required for proper codon-anticodon base pairing and maintenance of the reading frame [24,25]. Modifications in the D or T loops are required for tRNA stability and functional folding [26]. Due to their sequence differences, isoacceptors and isodecoders are unevenly modified resulting in tRNAs that carry the same aminoacid but have different physical and functional properties, providing unique characteristics to the cellular tRNA pool. Interestingly, the tRNA pool is not static. It is modulated during development and by environmental changes and sources of stress [14].

## 4. tRNA Fragments, a New Class of Small Noncoding RNAs

Although the function of tRNAs as adapter molecules is well known, recent studies have shown that they are a major source of small noncoding RNAs that, together with other small RNAs such as PIWI-interacting RNAs (piRNAs), ribosomal-derived small RNAs, small nucleolar RNAs (snRNAs), and microRNAs (miRNAs), play an active role in diverse biological processes such as RNA processing, post-transcriptional and post-translational regulation, cell proliferation and apoptosis, vesicle-mediated intercellular communication, and intergenerational inheritance [27,28,29,30,31,32,33,34,35,36,37,38,39,40,41,42]. Small RNAs derived from tRNAs are generated by enzymatic cleavage of the parental (precursor and mature) tRNAs [19,35,36,39,43] (Figure 1). A large variety of tRNA fragments exists which have been classified into seven groups based on their position within the parental tRNA sequence: 5′ or 3′ small tRNA fragments (tRFs) of ≤30 nucleotides that are generated from the 5′ or 3′ ends of the parental mature tRNA, internal tRNA fragments (i-tRFs) of varying lengths (16 to 33 nucleotides), 5′ or 3′ 33 nucleotide-long tRNA halves (tRHs) and 5′U-tRFs and tRF-1s derived from precursor tRNAs and containing part of the 5′ leader or 3′ trailer sequence, respectively [35] (Figure 1). The endonucleases Angiogenin, RNAse T2, Dicer, and RNaseZ/ELAC2 generate some but not all tRNA fragments [27,44,45,46], suggesting that other tRNA cleaving enzymes remain to be identified [47]. tRNA fragmentation may occur constitutively, as a result of differential expression of tRNA genes [48], or in response to stress conditions such as hypoxia, nutrient deprivation, inflammation, or oxidative stress [27,31,43,49,50,51]. Reduced tRNA modifications, as a consequence of mutations or impaired function of tRNA-modifying enzymes or increased demethylase activity, may also lead to tRNA fragmentation [52,53,54,55,56]. Dysregulated tRNA fragmentation has been detected in different diseases [35,36] including cancer [45,54,57,58], neurological disorders [52,59,60,61,62], metabolic disorders [30,31,53,55], and osteoarthritis [63]. 

Some classes of tRFs function in a miRNA-like manner leading to post-transcriptional mRNA silencing through RNA-induced silencing complex (RISC)-mediated RNA degradation [36,37,46,63,64,65]. Photoactivatable ribonucleoside-enhanced crosslinking and immunoprecipitation (PAR-CLIP) analysis identified tRFs bound to all four Argonaute proteins and PIWI proteins [66] which are the main effectors in miRNA-guided post-transcriptional RNA silencing [67,68]. Interestingly, it was demonstrated that many tRFs contain 5′ 7-8 nucleotide-long “seed sequences” that, similar to miRNAs, match target mRNAs in their 3′UTR regions leading to RISC-mediated mRNA degradation [37,63,69,70]. tRNA fragments were shown to regulate protein translation through a variety of mechanisms, e.g., by sequestering mRNAs and preventing translation [28], interacting with the a subunit of eukaryotic translation initiation factor 2 (eIF2a) in the actively translating ribosome [49], preventing the eIF4G/eIF4A complex from stabilizing capped mRNAs which blocks translation initiation [29], interacting with active polysomes [33], disrupting the stability of the mammalian multisynthetase complex which coordinates the assembly of the mature ribosome, directly inhibiting ribosomal function [38,40], or by competing with the mRNA for ribosome binding [41].

## 5. Dysregulated tRNA Metabolism in Diabetes

Alterations in tRNA biology have been associated with a large variety of human diseases including cancer, neurological, mitochondrial, and metabolic disorders [7,52,61,71,72,73,74]. Related to the latter, mutations in aaRSs [75,76,77,78,79,80,81,82] and intronic variants in the tRNA-modifying enzyme CDKAL1 [83,84,85] have been associated with increased risk for T2D and obesity. Mutations in mt-tRNA genes are associated with maternally inherited diabetes and deafness (MIDD) and insulin resistance [86,87,88,89,90] and mutations in the tRNA methyltransferase TRMT10A cause young onset diabetes and microcephaly [53,91,92,93,94,95,96]. Oxidative stress and iron deficiency have been associated with impaired tRNA modifications and diabetes [97,98], while diet-induced epididymal tRNA methylation and fragmentation were linked with intergenerational inheritance of metabolic traits [30,31,55]. Altogether, a large variety of alterations in tRNA metabolism may affect cell function resulting in diabetes development and not only genetic but also environmental factors contribute to this pathogenic process. Below we provide a comprehensive review of these aspects with a particular focus on T2D.

## 6. Genetic Factors Affecting tRNA Biology and Their Association with Impaired Glucose Metabolism 

### 6.1. Pathogenic Variants in CDKAL1 and TRMT10A 

GWAS and replication studies have shown that polymorphic variants within the *CDKAL1* locus are strongly associated with increased T2D risk and reduced insulin secretion in different populations [83,84,99,100,101,102,103,104,105,106,107,108,109,110]. All disease-associated SNPs are located in intron 5 of *CDKAL1* [83]. It has been proposed that these noncoding variants regulate CDKAL1 expression and splicing but the underlying mechanism is still unclear [83,100,111]. CDKAL1 is a tRNA methylthiotransferase associated with the endoplasmic reticulum (ER) through its hydrophobic C-terminal domain [84,112]. CDKAL1 catalyses the methylthiolation (ms^2^) of N^6^ threonyl carbamoyladenosine (t^6^A) at position 37 of cytosolic tRNA^Lys^
_(UUU)_ (Figure 1) [83,84,112,113]. Individuals carrying the *CDKAL1* risk allele have reduced CDKAL1-mediated ms^2^t^6^A37 tRNA^Lys^ modification [114], a modification necessary for the stabilization of the codon-anticodon interaction and contributing to the accurate translation of Lysine AAA and AAG codons during protein synthesis [84,115]. *CDKAL1* risk variant carriers have decreased first phase insulin secretion and normal insulin sensitivity [116,117]. In line with that, β-cell specific *Cdkal1* knockout (KO) mice showed marked glucose intolerance and impaired insulin secretion that was exacerbated by high fat feeding [83,84]. Isolated islets from these animals exhibited reduced first phase insulin secretion and impaired ATP production, something also seen in whole body *Cdkal1* KO mice [118]. During insulin biosynthesis insulin mRNA is initially translated as preproinsulin, a single chain biologically inactive polypeptide containing an N-terminal signal peptide that mediates its translocation into the ER lumen. Upon ER entry the signal peptide is cleaved giving rise to proinsulin, which is folded and stabilized in its tertiary structure via disulfide bond formation. Thereafter, proinsulin is cleaved into mature insulin (A and B chain) and C-peptide [119]. Human proinsulin and mouse proinsulin 1 contain two lysine residues, one of them located in the cleavage site between C-peptide and the A chain. Wei et al. demonstrated that the absence of *Cdkal1* in β-cells results in the misreading of Lys codons and erroneous aminoacid incorporation into proinsulin. This negatively affects proinsulin folding and inhibits its proteolytic cleavage resulting in misfolded proinsulin accumulation in the ER and ER stress [84]. Brambillasca et al. showed that *Cdkal1* silencing in rat insulinoma cells not only affects insulin but also reduces the expression of chromogranin A and islet cell autoantigen 512, which are present in insulin secretory granules [112] suggesting that the activity of this ER-localized tRNA-modifying enzyme is needed for efficient translation of certain secretory proteins [112]. In agreement with the phenotype of the β-cell specific *Cdkal1*-KO mice, *CDKAL1-*deficient human induced pluripotent stem cell (iPSC)-derived β-like cells showed defective glucose-stimulated insulin secretion and hypersensitivity to high glucose and FFA-induced ER stress and apoptosis, highlighting the importance of preserved CDKAL1-mediated tRNA^Lys^
_(UUU)_ modification for correct β-cell function [120]. 

Homozygous inactivating mutations in the *TRMT10A* gene cause young onset diabetes, microcephaly, intellectual disability, and epilepsy [91,92,93,94,95,96]. We were the first to report this association in three patients from a consanguineous family of Moroccan origin [91]. This original study was followed by five additional reports from independent research groups that identified nine additional patients from five unrelated families having biallelic nonsense, missense, or deletion mutations in *TRMT10A* and very similar clinical manifestations [92,93,94,95,96]. TRMT10A is a tRNA methyltransferase with major nuclear localization that catalyses guanosine 9 methylation (m^1^G_9_) in some cytosolic tRNAs (Figure 1) [53,91]. We and others have demonstrated that tRNA^Gln^_(UUG and CUG),_ tRNA^IniMeth^, tRNA^Arg^_,_ tRNA^Asn^_,_ tRNA^Trp^, and tRNA^Gly^_(CCC and GCC)_ are TRMT10A substrates [53,121,122,123]_._ In addition, in vitro methylation studies showed that mt-tRNA^Ile^, mt-tRNA^Leu^, and mt-tRNA^Tyr^ can also be methylated by TRMT10A [123]. TRMT10A is ubiquitously expressed but enriched in pancreatic islets and brain, which are the main tissues affected in the patients [91]. Using different rodent and human β-cell models of RNA interference-mediated TRMT10A deficiency, lymphoblasts and iPSC-derived β-like cells from nondiabetic individuals and TRMT10A diabetic patients, we showed that TRMT10A expression is induced by ER stress [91] and that TRMT10A-deficient β-cells are more sensitive to FFA-, ER stress-and high glucose-induced apoptosis [53,91]. The absence of TRMT10A results in reduced m^1^G_9_ methylation of TRMT10A tRNA targets and hypomethylated tRNA^Gln^ is prone to fragmentation. These 5′-tRNA^Gln^ fragments are key mediators of TRMT10A deficiency-induced β-cell demise [53]. Interestingly, TRMT10A-deficient cells showed enhanced oxidative stress and activation of the intrinsic (mitochondrial) pathway of apoptosis, characterized by increased splicing of the pro-apoptotic protein Bim [53]. The molecular mechanisms underlying 5′-tRNA^Gln^ fragment-associated β-cell demise still need to be unveiled. While being hypomodified, tRNA^IniMeth^ was not fragmented in patient cells suggesting that the absence of m^1^G_9_ modification does not equally affect all tRNAs [53]. Recently, it was shown that TRMT10A deficiency in the haploid human cell line HAP1 does not lead to tRNA^Gln^ fragmentation [124]. In these cells tRNA^IniMet^ hypomethylation was accompanied by a decrease in the abundance of this tRNA, something not observed in TRMT10A-deficient β-cells [53,124]. Altogether, these findings suggest that the outcome of impaired tRNA modification is probably tissue- and cell-specific. Whether additional TRMT10A tRNA substrates are fragmented in TRMT10A-deficient β-cells remains to be investigated. 

It has been recently demonstrated that, in HEK293T cells, TRMT10A indirectly regulates m^6^A mRNA methylation by interacting with Fat Mass and obesity-associated protein (FTO) [125]. FTO is a demethylase that removes m^6^A modification from certain mRNAs. This modification plays an important role in regulating mRNA stability since its presence targets the modified mRNAs to degradation and the presence of m^6^A in any position of most codons negatively affects translation-elongation dynamics [126,127]. It has been proposed that under normal conditions TRMT10A modulates the substrate selectivity of FTO and enhances its demethylase activity resulting in the stabilization of selected mRNAs. In the absence of TRMT10A, FTO activity is reduced with the concomitant increase in m^6^A mRNA modification, which results in mRNA degradation [125]. Interestingly, it was shown that in TRMT10A-deficient cells the degraded mRNAs were enriched in codons complementary to some TRMT10A tRNA targets (tRNA^Gln^_(UUG),_ tRNA^Arg^_(CCG)_, and tRNA^Thr^_(CGT)_ [125], suggesting the presence of a coordinated mechanism regulating gene expression through the interaction between mRNA and tRNA modifying enzymes [125]. This additional role of TRMT10A was shown to be independent of its tRNA methyltransferase activity. Variants in FTO have been associated with increased obesity and T2D risk suggesting that dysregulated FTO action or expression negatively affects glucose homeostasis in humans [128,129]. Whether FTO activity is modulated by TRMT10A in pancreatic β-cells, and its implications in gene expression regulation under TRMT10A deficiency still need to be studied. 

Interestingly, RNA sequencing data of FACS-purified pancreatic β-cells from four type 1 diabetes (T1D) and thirteen nondiabetic individuals showed a trend for a 60% decrease in TRMT10A mRNA expression in T1D β-cells [130]. Even if this reduction was not statistically significant, it suggests that TRMT10A deficiency might play a role in β-cell failure in T1D. 

### 6.2. Pathogenic Variants in aaRSs 

Accurate protein synthesis is very important to preserve cell function and survival and relies on proper aaRS-mediated tRNA aminoacylation. In mammalian cells cytosolic aaRSs are organized in a high molecular mass multisynthetase complex that comprises nine aaRSs (MARS1, IARS1, LARS1, EPRS1 (that is a bifunctional enzyme comprising EARS1 and PARS1), QARS1, DARS1 and KARS1) and three nonsynthetase accessory subunits (p43, p38, and p18) [131]. It has been proposed that the multisynthetase complex interacts with different components of the translational machinery contributing not only to tRNA charging but also to the delivery of charged tRNAs to the ribosome without diffusion into the cytoplasm. Translational fidelity is not always achieved. Under certain conditions the cells may synthesize mistranslated proteins in which an aminoacid is incorporated that differs from what is dictated by the genetic code [132]. This mistranslation may be done deliberately with beneficial effects, mainly when it occurs in response to stress conditions, e.g., nutrient deprivation or oxidative stress [133], but it may be toxic when it is due to aaRS mutations [132]. Indeed, mutations in several aaRS have been linked to human disease, such as myopathies and neuropathies [134]. The pathological consequences of reduced aaRS activity and consequent translational infidelity depend on the cell type and the extent of editing disruption [135]. In neurons, mutations affecting the editing domain of alanyl tRNA synthetase (Aars) compromises the proofreading activity of this enzyme during tRNA aminoacylation resulting in impaired translational fidelity, protein misfolding, and ER-stress [136]. Besides their conventional role in tRNA charging, several aaRS were shown to have additional noncanonical activities (e.g., regulation of apoptosis, cell proliferation, and extracellular matrix production) suggesting that impaired aaRS function may have wider implications than only alterations in protein translation [9].

A study in 7836 individuals from the Netherlands and Denmark identified a pathogenic missense H324Q variant in the *LARS2* gene, encoding mitochondrial leucyl-tRNA synthetase which catalyses the aminoacylation of mt-tRNA^Leu^_(UUR)_, as associated with T2D [75]. This initial association could, however, not be replicated in a much larger follow-up study [137]. The importance of preserved tRNA aminoacylation for proper β-cell function is illustrated by the association between impaired mt-tRNA^Leu^_(UUG)_ charging and MIDD [138,139,140]. This pathology, in which patients develop diabetes in adolescence or adulthood as a result of progressively impaired β-cell function [139] is mainly caused by a missense A3243G mutation in mt-tRNA^Leu^_(UUG)_ [86,139,141,142]. Experiments in cybrid cells, a system in which mutant mitochondria from patients are transferred into human cells lacking mtDNA, showed that the A3243G mutation causes defective mitochondrial protein translation, reduced oxidative phosphorylation, oxidative stress, and cell failure [143]. At the molecular level, the A3243G mutation, located in the D-loop of mt-tRNA^Leu^_(UUG)_, negatively affects translation of UUG-rich mitochondrial genes and reduces tRNA aminoacylation [138,139,140]. Related to the latter, *LARS2* overexpression in A3243G mt-tRNA^Leu^_(UUG)_ cybrid cells improved the aminoacylation of this mt-tRNA, increased the efficiency and fidelity of mitochondrial translation, and rescued mitochondrial dysfunction in the mutant cells [144]. These findings highlight the impact of impaired mt-tRNA aminoacylation on diabetes development. 

A genome wide association study (GWAS) performed in black South Africans identified one intronic and one 3′-UTR variant in the *WARS2* gene, encoding mitochondrial tryptophanyl-tRNA synthetase, as associated with increased hip circumference and waist-hip ratio, respectively [76,77]. Other single nucleotide polymorphisms (SNPs) in *WARS2* were found to be associated with waist-hip ratio, body mass index, and impaired fasting glucose in non-African populations [78,79,80,81]. Interestingly, a missense *Wars2* mutation in spontaneously hypertensive rats was shown to induce brown adipose tissue (BAT) dysfunction and promote visceral obesity [82]. BAT is responsible for nonshivering thermogenesis [145]. BAT plays an important role in lipid and glucose metabolism in rodents and possibly also in humans where BAT amount and activity are inversely correlated with age, glycemia, body mass index, and body fatness [146,147,148]. Mitochondrial dysfunction in BAT results in impaired thermogenesis and diet-induced obesity that can contribute to the development of T2D [145,146]. The *Wars2* missense mutation in rats negatively affected mitochondrial protein synthesis and mitochondrial function in BAT which was directly correlated with increased adiposity in these animals [82]. These findings highlight a potential novel mechanism of BAT dysfunction-mediated obesity associated with impaired mt-tRNA aminoacylation.

Altogether these data highlight the importance of preserved tRNA charging for proper cellular function. That negative consequences of impaired tRNA aminoacylation were seen essentially for mt-tRNAs is probably due to the limited number of mt-tRNA molecules, all of which are essential for proper mitochondrial protein synthesis and function. Tissues that functionally most depend on mitochondria such as skeletal muscle, BAT, or β-cells will be particularly sensitive to impaired mt-tRNA aminoacylation.

### 6.3. Regulation of aaRSs Expression by T2D Susceptibility Gene Variants

Kobiita et al. recently demonstrated that the T2D susceptibility gene JAZF1, the expression of which is reduced in pancreatic islets of T2D patients, is a transcriptional activator of several aaRS-encoding genes [149]. Interestingly, *Jazf1* deficiency in mouse β-cells resulted in reduced expression of Dars and Gars [149], the latter being the aaRS with highest expression in human pancreas [150]. Moreover, they also showed that Gars levels are reduced by 50% in db/db mouse islets, and Gars and Dars silencing sensitize pancreatic β-cells to ER stress-induced apoptosis. These findings suggest that some T2D gene variants increase T2D risk by altering aaRS expression. Kobiita et al. also found reduced JAZF1 expression in islets from two T1D patients when compared to four nondiabetic individuals, as well as in islets from eight-week-old NOD mice, a model of autoimmune diabetes [149]. RNA sequencing by Russell et al. of FACS-purified pancreatic β-cells from four T1D patients and thirteen nondiabetic individuals showed no changes in JAZF1 or GARS1 expression and a nonsignificant trend for reduced DARS1 levels in T1D β-cells [130]. Further studies are thus needed to assess whether JAZF1 and DARS1 expression is dysregulated in T1D and whether this is implicated in the pathogenesis of the disease.

### 6.4. Mutations in mt-tRNA Genes 

It has been largely demonstrated that mutations in mt-tRNA genes cause MIDD [86,87,88,89,90,139,141,151]. As mentioned above, most patients with mitochondrial diabetes bear a missense A3243G mutation in mt-tRNA^Leu^_(UUG)_ which reduces the taurinomethyluridine modification in position 34 of this tRNA resulting in impaired mitochondrial protein synthesis and mitochondrial dysfunction in tissues with high energy demand such as pancreatic β-cells, muscle, and neurons [86,87,138,139,141,142,143]. A missense A14692G mutation in mt-tRNA^Glu^ that reduces pseudouridination at position 55 of this tRNA resulting in mt-tRNA^Glu^ degradation was also reported to cause MIDD [89], and a T14709C mutation in the same mt-tRNA was identified as causal of syndromic forms of diabetes [152,153].

Three reports from the same research group have proposed that missense mutations in several mt-tRNAs are associated with polycystic ovary syndrome [90,154,155], a disease characterized by insulin resistance, hyperinsulinemia, hyperglycemia, and dyslipidemia [90,156,157]. The mt-tRNA variants were localized in evolutionary conserved sites and predicted to affect the secondary tRNA structure resulting in impaired tRNA modification, processing, or aminoacylation and impaired mitochondrial function [90]. Since mt-tRNA mutations affect mitochondria-rich tissues, the authors proposed that mitochondrial failure and increased oxidative stress in muscle may be the main contributors to insulin resistance in polycystic ovary syndrome [90]. The lack of traditional features of mt-tRNA mutations in these women and the homoplasmic nature of the mt-tRNA variants question the causal association between the identified variants and polycystic ovary syndrome and insulin resistance. The findings from this single research group need to be confirmed by multicentre studies [158].

## 7. Environmental Factors Affecting tRNA Aminoacylation, Modification, and Fragmentation 

### 7.1. Nongenetic Inhibition of tRNA-Modifying Enzymes 

Several studies reported the association between body iron excess and increased risk of T2D or gestational diabetes, in which the iron excess is thought to cause β-cell failure and insulin resistance [159,160,161,162,163,164,165]. The association of iron deficiency with diabetes is less clear. A large meta-analysis suggested that pregnant women with iron deficiency anemia have less risk of developing gestational diabetes [166]. In a recent study in mice iron deficiency impaired insulin secretion and caused diabetes [167]. Iron overload results in oxidative stress, β-cell dysfunction, and apoptosis [168,169], while iron deficiency causes mitochondrial failure [170], suggesting that iron levels have to be tightly regulated. Iron is necessary for the mitochondrial synthesis of iron-sulfur (Fe-S) clusters that are important cofactors for proper expression and activity of a large number of proteins of the respiratory chain and Krebs cycle as well as enzymes involved in DNA metabolism and tRNA modifications [170,171,172,173]. CDKAL1 and its mitochondrial homolog Cdk5 regulatory subunit-associated protein 1 (CDK5RAP1), which catalyses the ms^2^i^6^A37 modification in four mt-tRNAs (Figure 1), require two 4Fe-4S clusters and cysteine-persulfide (CysSSH) for their catalytic activity [97,98,113,173] (Figure 2). In mammals iron homeostasis is controlled by iron regulatory proteins 1 and 2 (Irp1 and Irp2) that regulate the expression of the transferrin receptor and two ferritin subunits (FtH1 and FtL1) that are involved in iron uptake and iron sequestration, respectively [174,175]. Santos and coworkers showed that *Irp2* KO mice develop diabetes as a consequence of iron deficiency [167]. These animals had fasting and fed hyperglycaemia as a consequence of reduced insulin and enhanced proinsulin secretion without changes in insulin sensitivity. Proinsulin accumulation in the ER induced ER stress [167]. Iron deficiency not only impaired mitochondrial protein function leading to reduced ATP production but also negatively affected CDKAL1 function, as shown by reduced ms^2^t^6^A37 tRNA^Lys^_(UUU)_ modification, reduced Lys incorporation into proinsulin and impaired proinsulin processing [167] (Figure 2). Iron supplementation rescued this phenotype in vitro and in vivo. This elegant study identified an iron-dependent modulation of CDKAL1 activity associated with diabetes development [167]. 

Oxidative stress may be another factor influencing the tRNA epigenome in pancreatic β-cells by affecting the activity of tRNA-modifying enzymes [73,97,98,113,170]. CysSHH is highly susceptible to oxidation [176] and oxidative stress affects the redox state and availability of 4Fe-4S clusters, suggesting that increased reactive oxygen species (ROS) derived from environmental insults or particular feeding conditions such as high fat diet (HFD) may lead to CDKAL1 and/or CDK5RAP1 inactivation [73,97,98,113,170] (Figure 2). Takahashi et al. showed that impaired tRNA ms^2^ modification caused by reduced CysSSH levels negatively affects insulin secretion in vitro and in vivo [98]. Indeed, the authors demonstrated that CysSSH acts as one of the sulphur donors in the methylthiolation reaction catalysed by CDKAL1. CysSSH deficiency, induced by silencing of cystathionine beta synthase and cystathionine gamma lyase (CTH) (both enzymes involved in CysSSH synthesis) or CTH inhibition results in reduced ms^2^ tRNA modification [98]. Interestingly, while chemical Cth inhibition in mice impaired glucose tolerance and reduced insulin secretion [98], *Cth* KO mice did not show metabolic alterations when fed normal chow. However, under high fat feeding these animals showed impaired glucose tolerance and significantly reduced insulin secretion compared to wildtype littermates [98]. These metabolic alterations were associated with reduced CysSSH levels and a concomitant decrease in ms^2^ modification in pancreatic islets, changes that were exacerbated by the HFD. Enhanced oxidative stress caused by HFD may downregulate the CysSSH and potentially 4Fe-4S cluster levels compromising CDKAL1 activity [98]. In agreement with these findings, we find that human islet exposure to the saturated FFA palmitate reduces ms^2^t^6^A37 tRNA^Lys^_(UUU)_ modification without affecting CDKAL1 expression (Figure 3). Wei et al. have shown that CDK5RAP1 activity is rapidly impaired by H_2_O_2_-mediated oxidative stress and this can be recovered by antioxidants [97]. Interestingly, in *Cdk5rap1* KO mouse embryonic fibroblasts the lack of the ms^2^i^6^A modification in mt-tRNAs leads to mitochondrial failure as a result of reduced mitochondrial protein synthesis affecting the respiratory complexes I, III, and IV [97]. Despite the mitochondrial dysfunction of Cdk5rap1-deficient cells, Cdk5rap1 KO mice develop normally and did not show any particular phenotypic defect when fed normal chow. When fed a low carbohydrate and high fat ketogenic diet, in which energy is generated primarily through fatty acid oxidation resulting in strong mitochondrial demand, these mice showed signs of mitochondrial failure in skeletal muscle and heart, two mitochondria-rich tissues. These mitochondrial defects were associated with reduced ms^2^i^6^A modification in mt-tRNAs [97]. The authors also showed that peripheral blood cells from patients bearing the A3243G mt-tRNA^Leu^_(UUG)_ mutation, which causes MIDD and mitochondrial encephalopathy, lactic acidosis, and stroke-like episodes, have reduced ms^2^ modification in the four CDK5RAP1 mt-tRNA substrates (mt-tRNA^Ser^_(UCN),_ mt-tRNA^Trp^, mt-tRNA^Phe^ and mt-tRNA^Tyr^), while no change was seen in the methythiolation of the CDKAL1 cytosolic substrate tRNA^Lys^_(UUU)_. The ms^2^ reduction in mt-tRNAs was the consequence of reduced CDK5RAP1 activity rather than reduced expression and it was directly correlated with the heteroplasmy level of mutant mt-DNA [97]. Since the mt-tRNA^Leu^_(UUG)_ mutation causes mitochondrial dysfunction and oxidative stress [143], the authors proposed that enhanced ROS levels may result in 4Fe-4S oxidation and CDK5RAP1 inactivation, pointing to oxidative stress-mediated CDK5RAP1 inactivation as an additional contributor to mitochondrial failure in MIDD and mitochondrial encephalopathy, lactic acidosis, and stroke-like episodes [97] (Figure 2).

### 7.2. Modulation of aaRS Activity by Nutrients and tRNA Fragments

Nutrient availability affects aminoacylation of specific tRNAs. It has been demonstrated that the aaRS LARS1 works as leucine sensor and upstream regulator of mTORC1 which is a key controller of catabolic and anabolic cell metabolism depending on nutrient availability [177]. Dysregulated mTORC1 activity has been implicated in the pathophysiology of different diseases including T2D [178]. In leucine-repleted conditions LARS1 promotes tRNA leucylation and mTORC1 activation which results in enhanced protein synthesis and cell growth [179]. Interestingly, Yoon et al. showed that the leucine-sensing function of LARS1 is modulated by glucose availability. Upon glucose starvation cell metabolism is shifted to energy preservation resulting in Unc-51 like autophagy activating kinase 1 (ULK1) activation. ULK1 phosphorylates LARS1 in S391 and S720 which impairs its ATP and leucine-binding capacity resulting in reduced tRNA leucylation, impaired mTORC1 signalling, reduced protein synthesis, and increased autophagy. The authors showed that LARS1 inactivation promotes leucin utilization into the tricarboxylic acid cycle for ATP production. This has been described as a protective mechanism under glucose starvation conditions that may contribute to prevent cell death by promoting leucine utilization for energy production [180]. Whether the activity of other aaRSs is also regulated by nutrients needs to be investigated. 

aaRS activity may further be modulated by tRFs [38]. A 19 nucleotide-long 5′-tRF derived from parental tRNA^Gln^ interacts with the multisynthetase complex in HeLa cells leading to modest changes in protein translation [38]. In yeast, three 3′-tRFs and one 5′-tRFs were shown to inhibit protein translation by interfering with the aminoacylation of their parental tRNA and global aminoacylation, probably through inhibition of ribosome-associated aaRSs [34]. As mentioned above, we showed that 5′-tRFs^Gln^ mediate pancreatic β-cell failure in diabetes caused by inactivating mutations in the tRNA-modifying enzyme TRMT10A [53]. No major changes in protein synthesis were seen in TRMT10A-deficient β-cells [91]. However, whether 5′-tRF^Gln^ modulates the expression of selected proteins and aaRSs activity remains to be investigated. 

### 7.3. Modulation of tRNA^[Ser]Sec^ Modifications by Selenium and Statins

Selenium is an essential trace element that plays an important role in human health by controlling metabolic and physiological functions through its incorporation into selenoproteins under the form of selenocysteine. In humans, 25 selenoproteins have been identified that are involved in antioxidant defence, oxidative protein folding in the ER, muscle development and function, immune function, and other functions [181,182,183,184]. Glutathione peroxidases and thioredoxin reductases are selenoproteins that play a key role in regulating cellular redox state. Selenocysteine is encoded by UGA codons. Decoding UGA as selenocysteine rather than stop requires a specific secondary structure in the selenoprotein mRNAs, several trans-acting factors, and the selenocysteine tRNA (tRNA^[Ser]Sec^) [185]. This tRNA, which is initially charged with serine and then transformed into selenocysteine, is crucial for selenoprotein synthesis [185]. tRNA^[Ser]Sec^ has only four posttranscriptional modifications. Among them, the TRIT1-mediated isopentenylation of adenosine at position 37 (i^6^A_37_) [186,187] is required for accurate decoding of the nonsense codon, and for subsequent mcm^5^Um methylation at position 34 of the tRNA that is required for proper tRNA function and selenoprotein synthesis [185,188]. Selenium deficiency is associated with impaired tRNA^[Ser]Sec^ methylation at position 34 [189]. Since dietary selenium availability regulates the expression and function of certain selenoproteins including glutathione peroxidases and thioredoxin reductases, it has been proposed that selenium supplementation could be beneficial for T2D patients by counteracting oxidative stress [190]. While optimal dietary selenium levels are beneficial, mouse experiments and observational cross-sectional studies in humans suggest that prolonged high selenium intake exceeding nutrient requirements is associated with T2D and insulin resistance through enhanced selenoprotein expression and reductive stress [184,190,191]. In mice, tRNA^[Ser]Sec^ deficiency in pancreatic β-cells and hypothalamus causes impaired glucose tolerance and insulin resistance [192]. Mice expressing a mutant version of tRNA^[Ser]Sec^, bearing an A to C/G transition at position 37 which hampers the i^6^A_37_ modification, have reduced selenoprotein formation [185,193] and enhanced oxidative stress, impaired glucose tolerance and insulin resistance [184]. Hypothalamic oxidative stress caused by tRNA^[Ser]Sec^ deficiency causes leptin resistance resulting in obesity [192]. Altogether, these findings point to the crucial role of tRNA^[Ser]Sec^ in selenoprotein synthesis and the need for fine-tuned dietary selenium levels to ensure adequate tRNA^[Ser]Sec^ posttranscriptional modification and function.

Cholesterol-lowering statins may reduce selenoprotein synthesis by impairing tRNA^[Ser]Sec^ i^6^A_37_ modification as a result of reduced mevalonate synthesis [193,194,195]. For the i^6^A_37_ modification, TRIT1 uses isopentenyl pyrophosphate, which is a direct metabolite of mevalonate. A reduction in mevalonate levels may therefore negatively impact the i^6^A_37_ modification [194,195]_._ It has indeed been demonstrated that lovastatin reduces selenoprotein synthesis in vitro [193]. Whether the increased risk for new onset T2D with statin treatment [196] is associated with reduced selenoprotein expression still needs to be investigated. 

### 7.4. tRNA Fragments in Intergenerational Inheritance of Metabolic Disorders

Another piece of evidence supporting the role of environmentally altered tRNAs metabolism in obesity and diabetes is the implication of tRNA fragments in the intergenerational inheritance of metabolic traits (Figure 4). Epidemiological and preclinical data suggest that paternal protein-poor or caloric-restricted diets as well as energy-dense diets may influence paternal heredity, predisposing offspring to obesity, insulin resistance, and T2D [197,198,199,200]. Female rats born from HFD-fed fathers have impaired glucose tolerance and reduced insulin secretion as a consequence of pancreatic β-cell dysfunction [201], while premating fasting of male mice was shown to influence the offspring’s glycemia [202]. Paternal diabetes induced by low-dose streptozotocin was shown to cause insulitis and impaired insulin secretion in the offspring [203], while paternal HFD-induced prediabetes caused filial glucose intolerance and insulin resistance that was transmitted up to the second generation [204]. In this latter study the pancreatic islets of the F1 and F2 offspring had altered expression of genes involved in glucose homeostasis and insulin secretion, together with altered cytosine methylation in loci thought to be upstream regulators of the modulated genes. Similar methylation changes were present in the sperm of the prediabetic fathers, and a large proportion of the differentially methylated genes identified in sperm were also differentially methylated in islets in the offspring, suggesting that epigenetic changes in sperm are the basis for transgenerational inheritance of diabetes risk [204]. All of these studies were performed after natural fecundation and did not exclude confounding factors such as molecules present in the seminal fluid or in the maternal reproductive tract at conception [205]. To address this and evaluate the real contribution of epigenetic inheritance via gametes on the susceptibility to develop T2D and obesity, Huypens and colleagues performed in vitro fertilization studies using gametes isolated from regular chow- and high fat-fed animals [205]. Sperm and oocytes isolated from these mice were used for in vitro fertilization using different parental combinations. The two cell embryos were then transferred into healthy chow-fed foster mothers, and the F1 generation from all the parental combinations were fed HFD and phenotypically characterized [205]. The progeny of HFD-fed mice was more susceptible to develop obesity and diabetes than the progeny of chow-fed mice, suggesting that epigenetic factors in gametes have an important role in the intergenerational transmission [205]; the mechanisms underlying the inheritance of these phenotypes were not studied. Similar findings were reported by Chen et al. who showed that F1 offspring of male HFD-fed mice developed impaired glucose tolerance and insulin resistance [30]. Injection of spermatozoid heads derived from male HFD-fed mice into oocytes from chow-fed females produced offspring that were glucose intolerant and insulin resistant compared to the ones born from sperm isolated from chow-fed animals [30]. Injection of purified total sperm RNA from both chow and HFD mice into normal zygotes phenocopied glucose intolerance but not insulin resistance in F1 offspring, suggesting the involvement of additional factors such as changes in histone modifications or DNA methylation [30]. In 2012, Peng and colleagues had demonstrated that tRHs were the most abundant type of small RNAs present in mouse sperm heads [206]. In good correlation with these initial findings, Chen et al. found that the sperm RNA was enriched in 30–40 nucleotide-long tRHs whose abundance was enhanced by high fat feeding [30]. Injection of purified tRHs from sperm of HFD-fed mice into normal zygotes resulted in glucose intolerant offspring, suggesting that 5′tRHs are probably responsible for the metabolic alterations seen in the progeny of HFD-fed mice [30]. The authors also showed that the sperm tRHs from HFD-fed animals had increased m^5^G and m^5^C modifications, the latter being introduced by the methyltransferase Dnmt2. In a follow up study, Zhang et al. showed that Dnmt2 depletion prevents the diet-induced methylation of the tRH fraction and alters the small RNA expression profile in sperm resulting in a loss of the dysmetabolic phenotype seen in the progeny of HFD-fed fathers [55]. Since in *Drosophila* Dnmt2-mediated methylation of tRNAs protected tRNA-derived fragments from degradation [207] it was proposed that the Dnmt2-mediated tRHs methylation was necessary for the functionality and stability of these tRFs and strictly required produce the metabolic phenotype in offspring of high fat-fed mice [55]. RNA sequencing of 8-cell embryos and blastocysts generated after injection of sperm-derived tRHs from HFD- or chow-fed mice showed that HFD tRHs reduce the expression of genes involved in metabolic regulation pathways and pancreatic β-cell function. Many of the downregulated genes had promoter regions that matched the sequences of 5′tRHs suggesting that sperm tRHs may affect metabolic gene expression via a transcriptional cascade to induce the observed metabolic disorders [30].

Cropley et al. provided additional proof of tRH-mediated inheritance of a metabolic phenotype up to the F2 generation in the absence of a dietary challenge [208]. In this study, a congenic rodent model of obesity and pre-diabetes was used in which the dominant obesogenic allele can be segregated away from the offspring. The authors showed that, when fed HFD, the offspring of genetically obese fathers develop marked obesity, glucose intolerance, and insulin resistance compared to the F1 generation of control mice on the same diet. This metabolic phenotype was transmitted to the F2 generation even when the F1 fathers were fed chow [208]. Analysis of the small RNA population isolated from sperm of the F1 mice showed important differences miRNA and tRHs populations. Indeed, F1 males from obese fathers presented enhanced 5′-tRH^Gly^_(CCC)_ and reduced 5′-tRH^Glu^_(CTC)_ levels. Even though further research is needed, the authors proposed that at least 5′-tRH^Glu^_(CTC)_ may act as a miRNA by binding to argonaute 2 protein leading to the silencing of selected genes [208]. In addition, de Castro Barbosa studied the metabolic phenotype of progeny of HFD-fed male rats as well as the sperm epigenome of the F0, F1, and F2 generations [209]. The F2 progeny of HFD-fed grandparents had lower insulin levels and mild glucose intolerance compared to the F2 from chow-fed grandparents. The sperm epigenome showed changes in DNA methylation, miRNAs expression, and 41 tRNA fragments, providing additional evidence that sperm tRNA fragments are in part responsible for the inherited metabolic phenotype [163]. 

Maternal HFD feeding results in obesogenic phenotypes and altered hedonic behaviours in offspring, a phenotype that persisted for three generations and was transmitted via the paternal lineage, independently from changes in sperm methylation [210,211]. Sarker et al. described increased sperm tRNA fragments in the F1 male progeny of HFD-fed females, indicating that mother overfeeding can alter tRNA metabolism of sperm in offspring. In keeping with the other studies, microinjection of sperm tRNA fragments (essentially tRHs) from the metabolically affected F1 progeny into normal zygotes reproduced the obesogenic eating phenotype (preference for energy dense diets) and addictive-like behaviours (preference for alcohol and amphetamines) seen in F1 male mice. Small RNA deep sequencing of F1 sperm revealed higher amounts 5′-tRHs in F1 mice from HFD-fed mothers compared to controls and identified 13 5′-tRHs that were differentially expressed between the two groups. Target prediction analysis for the differentially expressed tRHs showed that some may target genes associated with addictive patterns which could explain, at least in part, the hedonic and obesogenic traits of F2 offspring [212]. All these findings are summarized in (Table 1). Intergenerational inheritance in humans is difficult to study experimentally. There is increasing evidence, however, that diet affects both male fertility and offspring predisposition to metabolic diseases (reviewed and summarized in Natt et al. [213]). RNA sequencing showed that tRHs are the most frequent type of small RNA in human sperm [206,214]. A recent study revealed that human sperm RNA load is sensitive to changes in diet [215]. Natt and colleagues showed a significant increase in tRNA fragments, mostly i-tRFs and 3′-tRFs, after two weeks exposure of a high-sugar diet [215]. A previous study of sperm from lean vs. obese men also reported differences in small tRNA pools [216], while a second analysis of the same dataset discovered only differences in i-tRFs [215]. Intriguingly, the tRNA fragments that were increased in the study with a short exposure to high sugar were decreased in obese men. 

Taken together, these findings highlight the deleterious metabolic consequences of altered tRNA modifications and tRNA fragmentation, and the influence of nutrient excess or depletion that have a major impact on the heritability of metabolic and behavioural features. tRNA fragments alone, however, do not explain all the inherited metabolic alterations suggesting that dysregulated DNA methylation together with changes in other small RNA such as miRNAs also contribute to this process. Additional mechanistic studies are needed to understand the molecular consequences of enhanced tRNA fragmentation on metabolic phenoty.

## 8. Concluding Remarks

The in vitro, in vivo, and clinical data discussed above provide compelling evidence of the contribution of disturbed tRNA biology to diabetes development. The fact that not only genetic but also environmental factors may affect tRNA-modifying enzyme function, tRNA charging, and tRNA fragmentation adds complexity to this novel pathogenic mechanism contributing to impaired glucose metabolism. Human islet exposure to the saturated FFA palmitate, which causes β-cell dysfunction and death [217,218] impairs CDKAL1 function as evidenced by reduced tRNA^Lys^ ms^2^ modification (Figure 3). Further studies are needed to assess whether this is the result of palmitate-induced oxidative stress, and whether the impaired tRNA^Lys^ modification contributes to palmitate-induced human β-cell demise. The impact of nutrients or other environmental insults on the function of the tRNA-modifying enzymes TRMT10A or CDK5RAP1 remains to be investigated [73,97]. The deleterious effect of 5′-tRNA^Gln^ fragments generated in TRMT10A-deficient pancreatic β-cells [53], as well as the high fat feeding-induced tRHs that contribute to the nongenetic inheritance of metabolic traits [30,55,213,215] highlight the importance of tRNA fragmentation on cell functionality. The molecular mechanisms associating these tRNA fragments with insulin resistance and β-cell dysfunction are still largely unknown. Unveiling the implication of these small RNAs in the regulation of gene and protein expression in β-cells and peripheral tissues is of utmost importance since it may open the way for novel therapeutic opportunities in diabetes. Indeed, as for miRNAs, tRNA fragments can be antagonized by antisense oligonucleotides [52,53]. This approach needs to be further developed in order to target tRNA fragments in specific tissues in vivo. Whether impaired CDKAL1 or CDK5RAP1 activity or nutrient exposure results in tRNA fragment generation in β-cells has not been investigated. Studying tRNA fragmentation under these conditions may contribute to identifying additional players regulating gene and protein expression in T2D. The advances in high-throughput small RNA sequencing make it possible to examine tRNA fragmentation in different tissues or biological fluids using relatively small amounts of RNA [17]. In several cancers intra- and extracellular tRFs have been detected and are currently being considered as noninvasive biofluid markers to assess malignancy [219]. It will be of particular interest to examine whether circulating tRFs are increased in T2D [220].

## Figures and Tables

**Figure 1 ijms-22-00496-f001:**
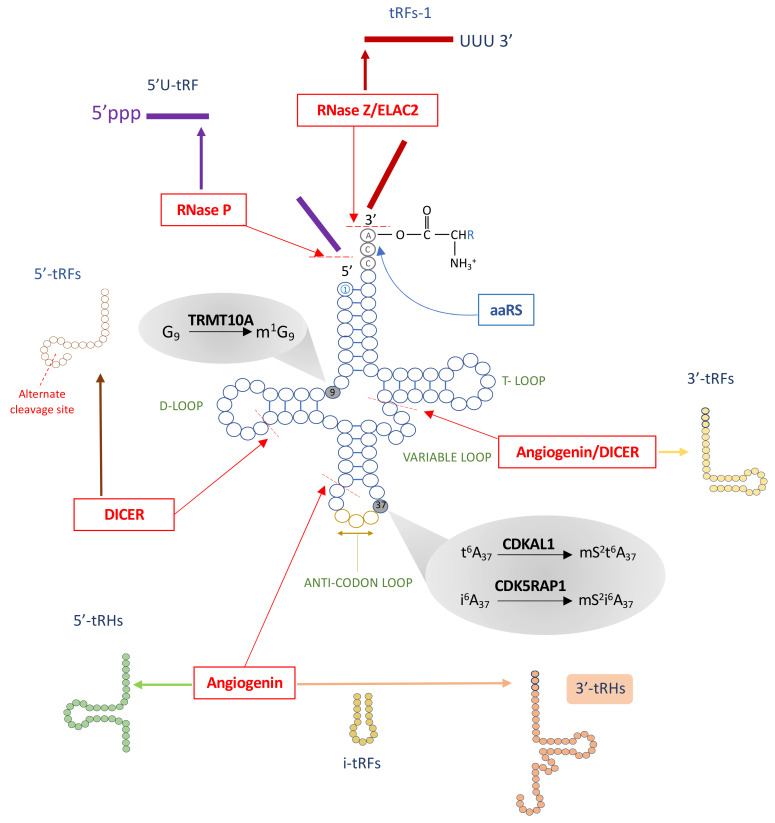
Cloverleaf 2D tRNA structure, tRNA modifications by TRMT10A, CDKAL1, and CDK5RAP1, tRNA aminoaylation, and biogenesis of tRNA fragments. Typical cloverleaf tRNAs structure of a cytosolic tRNA showing the D-loop, the T loop, the variable loop, and the anticodon loop. Mature tRNAs containing the CCA sequence in their 3′-end are covalently charged with their corresponding aminoacid a reaction catalysed by the aminoacyl-tRNA synthetases (aaRSs). tRNA molecules are postranscriptionally modified by tRNA modifying enzymes. TRMT10A is an S-adenosyl-L-methionine-dependent guanine N^1^-methyltransferase that catalyses the formation of N^1^-methylguanine at guanosine in position 9 of cytosolic tRNAs (m^1^ G_9_). Mutations in TRMT10A cause young onset diabetes and microcephaly. CDKAL1 catalyses the methylthiolation of N^6^-threonyl carbamoyladenosine (t^6^A), leading to the formation of 2-methylthio-N^6^-threonylcarbamoyladenosine (ms ^2^t^6^A_37_) at adenosine in position 37 of tRNA^Lys^_(UUU)_. Intronic polymorphisms in CDKAL1 increase T2D risk. CDK5RAP1 is a mitochondrial tRNA methylthiotransferase that catalyses the conversion of N^6^-(dimethylallyl)adenosine (i^6^A) to 2-methylthio-N^6^-(dimethylallyl)adenosine (ms ^2^i^6^A_37_) at adenosine in position 37 of different mt-tRNAs. Impaired CDK5RAP1 function has been associated with Maternal Inherited Diabetes and Deafness (MIDD) and Mitochondrial Encephalopathy, Lactic acidosis, and Stroke-like episodes. tRNA cleavage by Dicer, angiogenin, and other still unidentified endonucleases can give rise to tRNA fragments. Mature tRNAs are cleaved by DICER at the D-loop and T loop to produce 5′- and 3′- small tRNA fragments (5′-tRFs and 3′-tRFs), respectively. Angiogenin cleaves mature tRNA at the anticodon-loop to produce 5′ and 3′ tRNA halves (tRHs). Internal tRNA fragments (i-tRFs) comprise the anticodon- loop of the tRNA and are generated by unknown RNases. Immature (precursor) tRNAs having the 5′ leader (purple) and 3′ trailer (red) sequences are cleaved by RNaseP and RNase Z/ELAC2 generating 5′U-tRF and tRFs-1, respectively.

**Figure 2 ijms-22-00496-f002:**
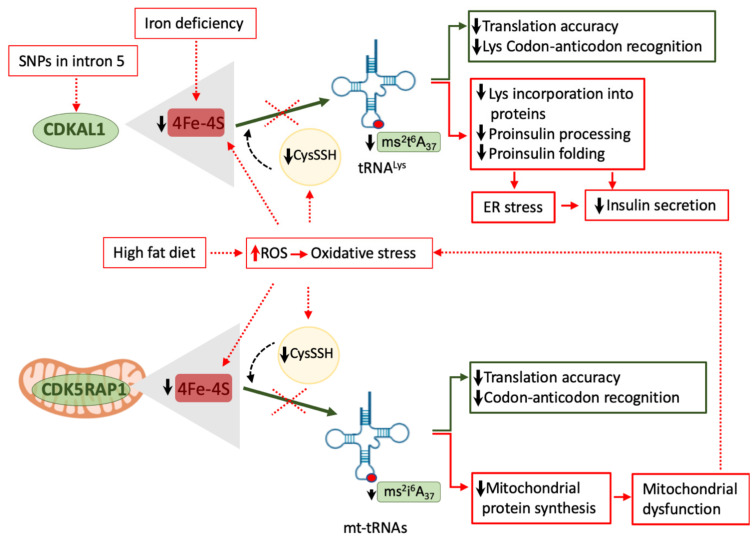
**Genetic and environmental inhibition of tRNA-modifying enzymes.** Genetic variants in CDKAL1 have been associated with increased T2D risk. Iron deficiency and oxidative stress (induced by HFD or mitochondrial dysfunction) impair the activity of both CDKAL1 and its mitochondrial homologue CDK5RAP1 by reducing the availability of 4Fe-4S clusters and CysSSH (both needed for their catalytic activity). Impaired CDKAL1 activity results in reduced ms^2^ modification in cytosolic tRNA^Lys^. This causes impaired tRNA^Lys^ incorporation into proteins (mainly proinsulin), reduced proinsulin processing, proinsulin accumulation in the ER that causes ER stress, and decreased insulin secretion. At the mitochondrial level, impaired CDK5RAP1 function leads to reduced ms^2^ modification in mt-tRNAs resulting in impaired mitochondrial protein synthesis. This results in mitochondrial dysfunction and oxidative stress that may amplify CDK5RAP1 failure.

**Figure 3 ijms-22-00496-f003:**
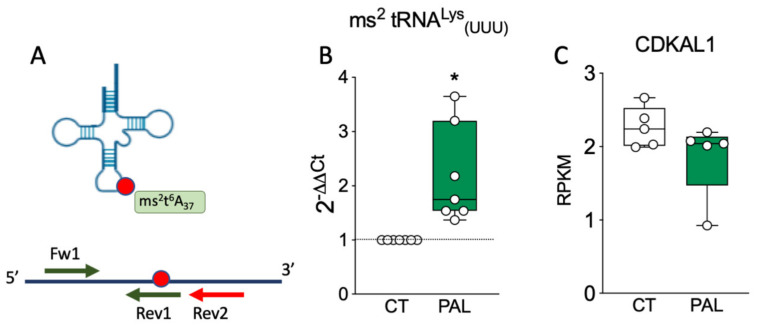
**The saturated free fatty acid palmitate impairs CDKAL1-mediated ms^2^ modification in cytosolic tRNA^Lys^_(UUU)_.** Human islets from seven nondiabetic organ donors were exposed or not (CT) for 48 h to 0.5 mM palmitate (PAL) in culture medium containing 1% BSA and no serum. Total tRNA was extracted and RNA^Lys^ ms^2^ was analysed by real-time PCR as described by Xie and colleagues [114]. (**A**) Representative scheme of the assay (image adapted from Xie et al. [114]). For assessing tRNA^Lys^ ms^2^ modification at A_37_, total RNA is reversed transcribed using Rev1 or Rev2 primers targeting human tRNA^Lys^. The cDNA obtained is then used as template for real-time PCR using the primer combinations (Fw1 + Rev1) or (Fw1 + Rev2). Xie et al. have shown that the ms^2^ modification blocks the reverse transcription. Thus, the presence of the ms^2^ modification impairs cDNA production when the Rev2 primer is used but it does not alter the reverse transcription with Rev1. (**B**) tRNA^Lys^ ms^2^ in human islets. The results (means ± SE) are expressed as 2^-ΔΔCt^. ΔCt = (Ct^Rev2-Fw1^ − Ct^Rev1-Fw1^). High 2^-ΔΔCt^ indicates reduced ms^2^ modification. This data indicates that PAL exposure reduces CDKAL1-mediated tRNA^Lys^ modification in human islets. (**C**) RNA sequencing data of CDKAL1 expression (expressed in RPKM, means ± SE, from Cnop et al.) of five human islet preparations exposed or not (CT) for 48h to PAL, showing that PAL exposure does not affect CDKAL1 expression. Data points represent independent human islet preparations.

**Figure 4 ijms-22-00496-f004:**
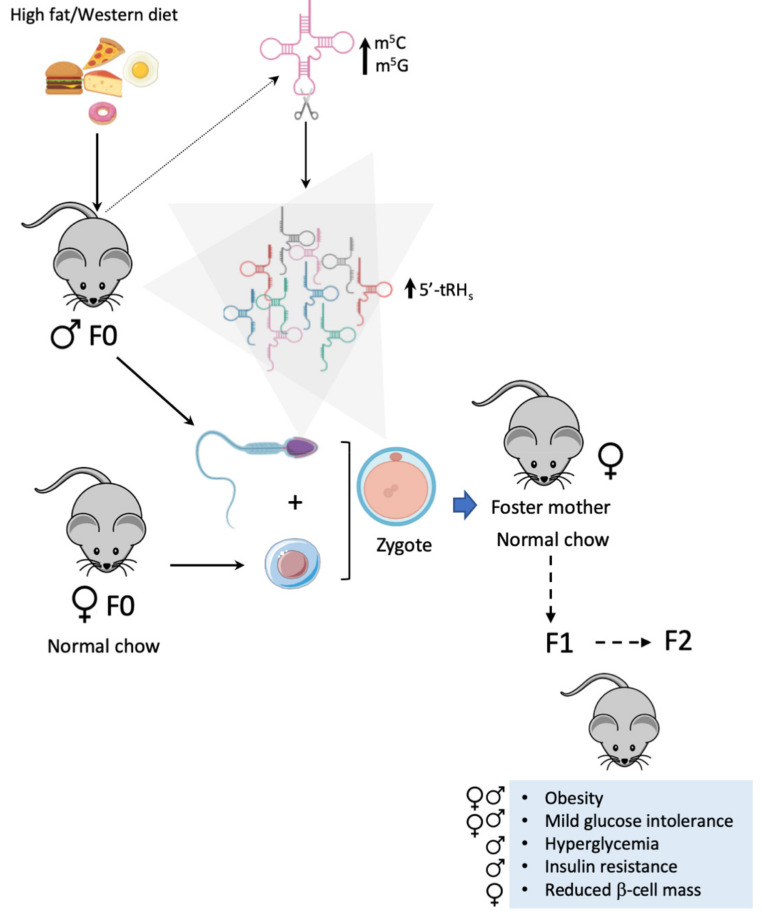
**Impact of hight fat diet-mediated tRNA fragmentation in intergenerational inheritance of metabolic traits.** High fat feeding in males leads to increased DNMT2-mediated m^5^C and m^5^G tRNA methylation and fragmentation in epidydimal and/or prostate acinar cells. The tRNA fragments generated (essentially 5′-tRHs) are transferred to sperm and transmitted to offspring upon fecundation affecting the metabolic health of F1 and F2 generations which show obesity, insulin resistance, mild glucose intolerance and reduced β-cell mass. Direct microinjection of sperm RNA fractions containing the 5′ tRHs into zygotes derived from normal chow-fed parents reproduced the metabolic phenotype in the offspring.

**Table 1 ijms-22-00496-t001:** **Paternal inheritance of metabolic traits.** This table recapitulates findings from all available studies reporting paternal inheritance of metabolic traits and their association with tRNA fragments present in sperm. HFD, high fat diet; IVF, in vitro fertilization; NC, normal chow, tRHs, tRNA halves.

	F0 (Diet or Genotype)		Offspring Phenotype			
Rodent Model and Breeding Method	F0 ♂	F0 ♀	F1	F2	Mechanism Unveiled	Reference Number
Rat/Natural mating	HFD	NC	**♀** β -cell dysfunction (glucose intolerance, ↓ insulin secretion)	-	642 differentially expressed genes in F1 islets. Hypomethylation of the Il13ra2 gene	[201]
Mouse/Natural mating	24 h premating fasting	NC	**♂****and****♀** ↓ non fasting glycemia	-	-	[202]
Mouse/Natural mating	Multiple low dose streptozotocin	NC	**♂** Insulitis ↓ insulin secretion	-	-	[203]
Mouse/Natural mating	HFD	NC	**♂****and****♀** Glucose intolerance Insulin resistance	-	Altered expression of genes involved in glucose homeostasis and insulin secretion in F1 islets. Altered cytosine DNA methylation in F1 islets and F0 sperm.	[204]
Mouse/IVF (using all gamete combinations)	6 weeks on HFD or NC	6 weeks on NC or HFD	**♂****and****♀** ↑ susceptivility to develop obesity and diabetes in progeny from HFD-fed F0 respect to progeny from NC-fed F0	-	-	[205]
Mouse/ IVF or zygote microinjection of total RNA or tRHs isolated from sperm from HFD-fed males	HFD	NC	**♂****and****♀** Glucose intolerance Insulin resistance (only after IVF)	-	↑ m^5^C and m^5^G in F0 sperm tRNAs. ↑ 5′-tRH in F0 sperm. ↓ expression of genes involved in metabolic regulation pathways in 8 cell F1 embryos.	[30,55]
Mouse/Natural mating	Congenic model of obesity and pre-diabetes A^vy^/a mouse) NC	NC (a/a)	**♂****and****♀** ↑ obesity, glucose intolerance and insulin resistance in HFD-fed offspring of A^vy^/a mice respect to control mice	**♂****and****♀** ↑ obesity, glucose intolerance and insulin resistance in the progeny of A^vy^/a grandfathers (even if F1 males were fed NC)	miRNA and tRH changes in F1 sperm from A^vy^/a fathers respect to F1 from control fathers: ↑ 5′-tRH^Gly^_(CCC)_ and ↓ 5′-tRH^Glu^_(CTC)_.5′-tRHs^Glu^_(CTC)_ binds to Argonaute and may act as miRNA regulating gene expression.	[208]
Rat/ Natural mating	HFD for 12 weeks	NC	**♀**↓ β-cell mass Glucose intolerance	**♀** ↓ insulin secretion Glucose intolerance	DNA methylation changes in sperm from F0 and F1. Differential expression of miRNAs and 41 tRNA fragments.	[209]
Mouse/ Natural mating in F0 (the F2 generation was obtained by microinjection of total RNA or 5-tRHs from F1 sperm into normal zygotes)	NC	HFD for 9 weeks	**♂****and ♀** ↑ weight gain ↑ blood glucose ↑ insulin secretion ↑ obesogenic eating phenotype ↑addictive-like behaviour		Identification of 13 differentially expressed 5′-tRHs in sperm from F1 derived from HFD-mothers.	[210]

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
