# Peer review of "tRNA Biology in the Pathogenesis of Diabetes: Role of Genetic and Environmental Factors"

_ijms, 2021, doi:10.3390/ijms22020496_

Round 1
Reviewer 1 Report
The manuscript reviewed tRNA biology in the pathogenesis of monogenic and type 2 diabetes and focused on the effects in genetic and environmental factors. The manuscript was well-prepared without self-promotion and the content they described match the title adequately. I have few comments as follow.
- In the figure 1, tRNA aminoacylation, modifications and fragmentation could be illustrated in one figure simultaneously to facilitate readability for unexperienced readers in this field.
- The content of the manuscript is too long. They could shorten some parts, such as section “7.3. tRNA fragments in intergenerational inheritance of metabolic disorders”. Experimental design might not be described so detail.
- The structure difference between Preproinsulin, Proinsulin and insulin was not well-addressed. Thus, it might be difficult for readers to understand clearly how tRNA defects links to accurate insulin process.
- Table 1 was not well-arranged, it should be improved.
- In the figure 4, line 495, “-cell mass” should be corrected as “b-cell mass”.
Author Response
Response to Reviewer 1 (Please see attachment)

Reviewer 2 Report
This is a comprehensive, well-written review on tRNA biology disturbances affecting energy metabolism and leading to type 2 diabetes . The authors provide a fascinating journey into the tRNA world to the reader, adding significant insights to the current literature. Below some comments that may improve further the manuscript. If Gars and Dars silencing sensitizes b-cells to ER-stress-induced apoptosis, have they been also connected to type 1 diabetes? Although the focus of the review is on T2D, it would be interesting to mention T1D in some cases as well, particularly as so many of the genetic-driven dysregulation of tRNA function leads to deficiencies involving insulin synthesis and/or secretion in b-cells. Mutations on the tRNA for selenocysteine have been shown to lead to glucose intolerance and a type 2 diabetes phenotype in mice (PMID: 21194350). As this mutation can be connected to dietary selenium and its role in selenoprotein synthesis, it should be mentioned as well in Section 7. The paradoxical role of statins, the most commonly prescribed drug to treat obesity-driven dyslipidemia, in decreasing hepatic isopentenylation of tRNAs should also be discussed in the manuscript. Minor comments: -Acronym tRF and tRH should be defined at its 1st appearance, in the Fig 1 legend. -Font format is not homogeneous throughout the text. -L.185: TRMT10A, not TRMT01A. -L.276: "Tissues that functionally MOST depend" -L.495: missing "beta"-cell mass.Author Response
Response to Reviewer 2 (Please see attachment)
